# Targeting Both Autophagy and Immunotherapy in Breast Cancer Treatment

**DOI:** 10.3390/metabo12100966

**Published:** 2022-10-12

**Authors:** Spyridon Giannopoulos, Cansu Cimen Bozkus, Eleni Zografos, Aikaterini Athanasiou, Ann Marie Bongiovanni, Georgios Doulaveris, Chris N. Bakoyiannis, Georgios E. Theodoropoulos, Georgios C. Zografos, Steven S. Witkin, Theofano Orfanelli

**Affiliations:** 1Department of Surgery, Indiana University School of Medicine, Indianapolis, IN 46202, USA; 2Department of Hematology and Medical Oncology, Icahn School of Medicine at Mount Sinai Hospital, New York, NY 10029, USA; 3Department of Clinical Therapeutics, Alexandra Hospital, National and Kapodistrian University of Athens, 15772 Athens, Greece; 4Department of Obstetrics and Gynecology, Weill Cornell Medicine, New York, NY 10021, USA; 5First Department of Surgery, Division of Vascular Surgery, Laikon General Hospital, National Kapodistrian University of Athens, 15772 Athens, Greece; 6First Department of Propaedeutic Surgery, Hippocration General Hospital, National and Kapodistrian University of Athens, 15772 Athens, Greece

**Keywords:** breast cancer, autophagy, immunotherapy, immunoregulation, immune-checkpoint inhibitors

## Abstract

As clinical efforts towards breast-conserving therapy and prolonging survival of those with metastatic breast cancer increase, innovative approaches with the use of biologics are on the rise. Two areas of current focus are cancer immunotherapy and autophagy, both of which have been well-studied independently but have recently been shown to have intertwining roles in cancer. An increased understanding of their interactions could provide new insights that result in novel diagnostic, prognostic, and therapeutic strategies. In this breast cancer-focused review, we explore the interactions between autophagy and two clinically relevant immune checkpoint pathways; the programmed cell death-1 receptor with its ligand (PD-L1)/PD-1 and the cytotoxic T-lymphocyte-associated protein 4 (CTLA-4)/CD80 and CD86 (B7-1 and B7-2). Furthermore, we discuss emerging preclinical and clinical data supporting targeting both immunotherapy and autophagy pathway manipulation as a promising approach in the treatment of breast cancer.

## 1. Introduction

Breast cancer is a leading cause of cancer deaths among women [1]. According to the American Cancer Society, a woman living in the United States has a 12.4%, or a one in eight, lifetime risk of being diagnosed with breast cancer [1]. Estimated new cases per year can approach one million, making breast cancer the most common malignancy in women, and up to 18% of all of the cancers found in women [2]. Treatment of breast cancer is approached from a variety of modalities, including surgery, radiotherapy, and chemotherapy. For the most part, treatment of early-stage breast cancers will involve a primary surgery with lumpectomy, followed by adjuvant radiation, which allows for better breast conservation. Radiation therapy can eradicate tumor deposits that remain in the breast tissue following surgery, thus reducing the risk of recurrence [3]. Systemic therapy approaches vary greatly depending on the tumor characteristics and the stage of the breast cancer being treated and can be adjuvant or neoadjuvant.

Immunotherapy, which targets immune pathways relevant to tumor survival and progression, has emerged as an innovative therapeutic approach in oncology. In recent years, therapies targeting “immune checkpoints”, or specific components of these pathways, have been shown to impede tumor cell growth in a number of malignancies without harming healthy tissue. Interfering with mechanisms that inhibit immune activation can reinvigorate anti-cancer cell immune responses, potentially leading to the complete elimination of the tumors. Breast cancer has traditionally been considered to be weakly immunogenic with a lower mutational load than other tumor types [4]. However, subtypes, particularly the “triple-negative” breast cancer (TNBC) that lack expression of estrogen receptors (ER), progesterone receptors (PR), and HER2 receptors (a member of the human epidermal growth factor receptor family), have been shown to be susceptible to immunotherapeutic agents that block immunosuppressive receptors [5]. A growing area of research for new treatments in breast cancer focuses on immunotherapy. Biologic drugs such as pembrolizumab and atezolizumab are specifically being investigated in relation to breast cancer [6]. Trials considering their use in conjunction with chemotherapy are currently under investigation [7].

Autophagy, a fundamental process present in almost all cells, maintains intracellular homeostasis and viability by removing non-functional organelles, degraded macromolecules, and intracellular microorganisms from the cytoplasm. These components become sequestered within a structure called an autophagosome. Subsequent fusion with a lysosome degrades the enclosed macromolecules and returns the building blocks to the cytoplasm for reutilization [8,9]. Autophagy has also been identified as an intriguing, emergent mechanism promoting tumor survival, metastasis, chemoresistance, and immunosurveillance in breast cancer [10]. Drugs that function as autophagy inducers or inhibitors, such as rapamycin, chloroquine, and hydroxychloroquine, have been studied as promising therapeutic strategies in breast cancer either as single agents or in conjunction with conventional chemotherapy or hormonal treatment [11,12].

This review focuses on examining the crosstalk between two of the most well-studied immune checkpoint molecules and the autophagy pathway in breast malignancy. We also discuss the data on combinations of immunotherapy and autophagy in breast cancer that may shortly enter clinical assessment. We propose mechanisms of how treatments that inhibit the activity of checkpoint signaling, combined with treatments that either inhibit or promote autophagy, may be especially beneficial in the treatment of breast cancer.

## 2. Autophagy and Breast Cancer

Autophagy is a word derived from the Greek language meaning “self-eating”, first named by Christian de Duve, more than 50 years ago. This highly regulated catabolic intracellular process plays an essential housekeeping role in removing toxic byproducts of metabolism, and eliminating misfolded or degraded protein aggregates, non-functional or aged organelles, and microbial components [13]. In times of stress, such as nutrient or oxygen deprivation, infection, or exposure to non-physiological conditions, the level of autophagy increases to enhance cell survival [14]. Autophagic dysregulation that results in an impairment of intracellular homeostasis and metabolism has been associated with the pathogenesis of different diseases, including cancer [15]. Interestingly, autophagy has a dual role in malignancies, acting both in an inhibitory manner by preventing the aberrant accumulation of dysfunctional cytosolic proteins and organelles in healthy cells, and, conversely, by promoting continued cell survival in malignantly transformed cells [16].

One of the initial studies highlighting the inverse relationship between autophagic activity and malignant potential was published by Qu et al. [17]. These researchers documented the essential role of autophagy in cell-growth control and tumor suppression. They provided evidence in a mouse model that the deletion of BECN1, a gene coding for a protein essential for autophagy induction, promoted tumorigenesis and led to a significantly higher frequency of spontaneous malignancies. Recent concurrent studies have confirmed that autophagy defects in healthy cells are strongly correlated with susceptibility to metabolic stress, genomic damage, and carcinogenesis [18,19]. Conversely, accumulating evidence also highlights the protective role of autophagy in promoting the viability of malignant cells, which often display higher basal autophagic activity than their non-malignant counterparts [20]. Specifically, once tumors are established and subjected to environmental stresses, such as hypoxia and nutrient insufficiency in poorly vascularized regions, autophagy allows the transformed cells to recycle cellular components, receive adequate nutrition, and survive [21]. Therefore, the up-regulation of autophagy helps maintain cancer cells’ viability and rapid proliferation and satisfies the increasing demand for components essential for macromolecular synthesis [22].

In this context, preclinical research studies have demonstrated that inhibition of autophagy can decrease tumor resistance to chemotherapy and improve anti-tumor responses in cancer patients, including those with breast cancer (Table 1) [23,24]. Anti-endocrine therapy with tamoxifen reportedly inhibits autophagy induction in estrogen receptor-positive breast cancer cells [25]. Autophagy inhibitors, such as the anti-malarial drug, chloroquine (CQ), have been shown to alter estrogen responsiveness in endocrine-resistant breast lesions [26]. Additionally, in postmenopausal women with hormone-dependent breast tumors, inhibition of autophagy may reverse the acquired resistance to the aromatase inhibitor, Exemestane. This re-sensitization of breast cancer cells to Exemestane occurs by apoptosis induction, cell cycle deregulation, and the inhibition of cell survival pathways [27]. Mammalian target of rapamycin (mTOR) is a central negative regulator of autophagy. Everolimus, an mTOR inhibitor that has recently been developed for the treatment of advanced hormone receptor positive, HER2-negative breast cancer, induces cell cycle arrest [28]. The induction of autophagy in aromatase inhibitor-resistant breast cancer cells enhances tumor cell survival and thereby contributes to Everolimus insensitivity [29,30]. Importantly, it has been demonstrated that autophagy is activated in breast cancer cells in response to Palbociclib, a CDK 4/6 inhibitor. Therefore, a therapeutic strategy that combines autophagy inhibition and CDK4/6 blockade would significantly enhance the sensitivity of breast cancer cells to treatment [31]. Consequently, it is becoming increasingly apparent that autophagy mediates tumor cell resistance to several agents utilized to treat breast cancer [32], emphasizing the potential value of autophagy inhibition as a parallel co-target for novel breast cancer therapeutics.

The above-mentioned evidence has led to the initiation of several clinical trials investigating the use of autophagy inhibitors, such as chloroquine (CQ) and hydroxychloroquine (HCQ), alone or in combination with other therapies, for breast cancer [24]. Arnaout et al. recently published the outcome of a Phase-II, randomized, double-blind clinical trial (NCT02333890) evaluating the effects of breast cancer treatment with chloroquine in a preoperative setting. The results were underwhelming since no significant changes in proliferative response indices were observed, while CQ toxicity was noteworthy [33]. Therefore, despite the various preclinical findings, summarized in Table 2, further ongoing studies are necessary to clarify the potential clinical effectiveness of autophagy inhibitors in the treatment of breast cancer.

## 3. Immune Checkpoint Molecules and Breast Cancer

The human immune system performs protective functions through complex pathways that strike a delicate balance between inducing effector functions and maintaining self-tolerance. Many of these pathways are mediated by T lymphocytes, and are regulated by the checkpoint system [34]. The fully efficient activation of T lymphocytes involves three signals. The first signal is recognition by a receptor on the membrane of T cells of a foreign antigen-major histocompatibility complex (MHC) protein complex present on the surface of antigen-presenting cells (APCs). The second signal, required for T cell activation, is an antigen-independent costimulatory signal provided by the engagement of CD80 or CD86 ligands on the APCs with the CD28 receptor on the T cell surface. The final signal is provided by the induction of interleukin-2 (IL-2) production that stimulates activated T cell proliferation. Following this sequence of events, antigen-specific cytotoxic T lymphocytes (CTLs) are generated that recognize and kill cells that express the MHC-bound antigen [35].

T cell activation is tightly regulated by checkpoint inhibitors to prevent their induction in response to self-antigens, which would initiate self-tissue destruction and development of autoimmune disease. CTLA-4 is a member of the immunoglobulin superfamily originally discovered more than 30 years ago. CTLA-4 mediates immunosuppression by competing with a higher affinity for binding to the B7 (CD80/CD86) co-stimulatory molecules on the surface of APCs [36]. Upon antigen presentation, this negates CD28 binding to its B7 ligands, thus diminishing CD28-mediated signaling, the release of IL-2, and decreased T cell proliferation [37]. PD-1 is a transmembrane receptor protein transcriptionally induced in activated T cells, B cells, and myeloid cells. It belongs to the CD28/CTLA-4 immunoglobulin superfamily [38]. Programmed cell death protein-1 (PD-1) was first identified in the early 1990s as an inhibitory molecule that regulates the late phase immune response in peripheral tissues; PD-1 loss resulted in an impairment of peripheral tolerance [38]. A few years later, Freeman and colleagues identified PD-L1 as the ligand for PD-1 and showed that engagement of PD-1 by PD-L1 led to the inhibition of T cell receptor-mediated lymphocyte proliferation and cytokine secretion [39].

Some cancers have evolved mechanisms to take advantage of these immune regulatory mechanisms by enhancing the overexpression of checkpoint inhibitors to inhibit CTL activation. By manipulating the immune checkpoint system, the tumors avoid becoming targets of the host’s immune system; the resulting immune tolerance enables their proliferation. The characterization of checkpoint signaling pathways and their manipulation has been a growing area of research for biologic therapy, especially in solid tumor cancers [34,40]. In the last decade, the generation of compounds, such as anti-CTLA-4 and anti-PD-1 monoclonal antibodies, that negate the activity of the corresponding immune checkpoint inhibitors has played a crucial role in development of novel therapies for a multitude of malignancies. In 1996, the first efforts in applying new knowledge of checkpoint inhibitors to cancer therapeutics were undertaken by Allison and colleagues [41]. Their research focused on blocking the inhibitory effects of CTLA-4 with a monoclonal antibody, leading to enhanced anti-tumor immune responses. After this breakthrough, which revealed the anti-tumor potential of immune checkpoint blockade, hallmark clinical trials resulted in development of the anti-CTLA-4 monoclonal antibody, Ipilimumab, which was approved for treatment of melanoma in 2011 [42,43,44]. The discovery of ipilimumab and its encouraging results in increasing overall survival of melanoma patients stimulated interest and paved the way for utilizing previous pre-clinical research on additional inhibitory molecules relevant to T cell function [38,45,46]. These efforts eventually led to the generation of monoclonal antibodies targeting PD-1 receptors.

In vivo and in vitro experiments detected the abundant expression of the PD-L1 immunoinhibitory protein in the tumor microenvironment in various human cancers but not in normal tissues [47]. Further evidence indicated that activated T cell interaction with tumor-associated PD-L1 led to programmed cell death. Thus, the PD-1/PD-L1 pathway functioned as a resistance mechanism by which tumors escaped endogenous immune destruction. Collectively, pre-clinical findings nurtured the idea of facilitating cancer immunotherapy by blocking PD-1 and PD-L1 [48,49]. The first clinical trial was launched in 2006 [50], and since then, six immune checkpoint inhibitors for the PD-1/PD-L1 pathway have been approved, either targeting PD-1 (pembrolizumab, NCT01295827; nivolumab, CheckMate Clinical Trials; cemiplimab, NCT02760498- NCT02383212) or PD-L1 (atezolizumab, NCT01693562; avelumab, NCT01772004; durvalumab, NCT01693562).

In breast cancer, the use of immunotherapy has been hindered by the immunosuppressive characteristics of this malignancy. Specifically, breast tumors have not been traditionally considered highly immunogenic since most exhibit poor lymphocyte infiltration, a low mutational burden, and a limited response rate to anti-PD-1/L1 monotherapy [51]. Despite these non-encouraging factors, attempts to exploit the immune system for anti-tumor responses have rigorously proceeded, especially in cases of TNBC. In 2019, an anti-PD-L1 monoclonal antibody (atezolizumab) was approved to treat patients with unresectable locally advanced or metastatic TNBC, not as a single-agent but in combination with nanoparticle albumin-bound (nab)-paclitaxel [52]. According to the results of the IMpassion130 trial (NCT02425891), the combination of atezolizumab with nab-paclitaxel as a first-line treatment in metastatic TNBC led to significantly prolonged progression-free survival in both the intent-to-treat population and the subgroup of patients who were positive for PD-L1 expression (≥1%) on immune infiltrates. This first approval has increased enthusiasm for further preclinical studies and clinical trials in the field of breast immuno-oncology, both with an investigation of the efficacy of immune checkpoint blockade monotherapy, and in the combination of immuno-oncology agents with existing regimens in breast cancer (Table 3) [53]. However, breast cancer is a complex disease, with each molecular subtype exhibiting a heterogeneity of response to checkpoint blockade therapy, as well as varying PD-L1 and CTLA-4 expression, nonsynonymous tumor mutational burden, and expression of pro-inflammatory cytokines [54,55,56]. Consequently, strategies and targeted interventions are still to be optimized to enhance the immune response and render heterogenous breast cancer tumors more responsive to immunotherapy.

## 4. Immune Checkpoint Molecules and Autophagy in Breast Cancer

### 4.1. PD1/PD-L1 and Autophagy in Breast Cancer

As mentioned above, induction of autophagy may have contradictory roles in both the development of anti-tumor immunity as well as enhancing the survival of malignantly transformed cells [57]. Autophagy induction can facilitate survival of the cancer cell during non-physiological conditions such as aberrant protein expression and nutrient deprivation, facilitate resistance of anti-cancer drugs, and thereby promote tumor progression and metastasis [58,59,60,61]. As such, several investigators have suggested that inhibition of autophagy can promote tumor cell destruction and is a possible target for breast cancer treatment [62,63]. Conversely, others suggest that disrupting autophagy in breast tumors would accelerate tumorigenesis [63,64].

There is evidence that the autophagy pathway is affected by the PD-L1 axis. Specifically, Zhang et al. demonstrated that inhibition of an endogenous autophagy inhibitor, mTORC1/2, resulted in a decrease in PD-L1 levels in human non-small cell lung cancer (NSCLC) cells [65]. This change in the PD-L1 levels was thought to be due to mTORC1 inhibition at the post-translational level. According to these findings, the mTOR pathway seems to mediate the expression of PD-L1. Therefore, cancer cells that express high levels of PD-L1 have a lower level of autophagy [66,67]. This intrinsic regulation of PD-L1 by autophagy was investigated by Wang et al., who showed that the autophagy blockade increased PD-L1 levels in gastric cancer cells [68]. Concomitantly, PD1 engagement in T cells inhibited their ability to respond to tumor antigens and thus inhibited anti-tumor immunity. Novel preclinical evidence supports the potential role of autophagy in tumor immunotherapy for treating ovarian cancer and melanoma. Specifically, enhanced PD-L1 signals sensitized tumor cells to the autophagy inhibitor chloroquine in vitro, through several overlapping mechanisms [67]. Therefore, cancer cells with high-level expression of the PD-L1 receptor could be a potential target for autophagy inhibitors compared to cells that weakly express PD-L1 [69].

However, some studies have found that these regulatory interactions between autophagy and immunoregulatory mechanisms function differently in breast cancer. Notably, autophagy-mediated PD-L1 degradation in Sigma1-expressing TNBC (sigma1 is a scaffolding protein involved in protein homeostasis in the endoplasmic reticulum (ER); it supports the increased protein synthesis demand associated with tumor growth) [70,71,72] and androgen-dependent prostate cancer cell lines contributed to tumor regression [73].

### 4.2. CTLA-4 and Autophagy in Breast Cancer

The CTLA-4 checkpoint inhibitor system and its roles in autophagy and cancer immunity have been mostly evaluated in melanoma. Studies have shown that an anti-CTLA-4 antibody has the potential to promote the host immune response. This effect was studied in mouse tumor models, leading to an FDA-approved anti-CTLA-4 antibody, ipilimumab, to treat melanoma [40]. The PI3K/AKT/mTOR pathway is a signal transduction pathway inhibiting autophagy induction that is abnormally activated in many tumors and plays a role in cancer development. Activation of this pathway by CTLA-4 enhances T cell survival and inhibits autophagy by reducing transcription of the protein, microtubule-associated protein 1 light chain 3 β (LC-3β), that is required for autophagosome formation [74,75]. In this context, the antibody-mediated blockade of CTLA-4 downregulates the activation of PI3K/AKT/mTOR by reducing the expression of autophagy-related proteins. We can speculate that an interaction similar to that observed between autophagy and PD-L1 may occur between autophagy and the CTLA-4 checkpoint receptor and its ligand; further studies in breast cancer investigating this possible connection are needed. If data from studies on other cancers can be extrapolated to breast cancer, the findings might be relevant to this malignancy.

Notably, autophagy suppression in human melanomas resistant to CTLA-4 inhibitors, but not to PD-1, has been associated with the expression of cancer germline antigens. This suggests that autophagy suppression may play a role in promoting resistance to CTLA-4 inhibitors. Therefore, the combination of autophagy induction and a CTLA-4 blockade could have a synergistic anti-tumor effect [76]. Furthermore, loss of expression of PTEN—a gene that encodes a phosphatase involved in the development of many types of cancer, including breast cancer [77]—in preclinical murine models of melanoma inhibited autophagy and decreased T cell-mediated cytotoxicity. Treatment with a selective phosphatidylinositol-4,5-bisphosphate 3-kinase (PI3Kβ) inhibitor improved the efficacy of both anti-PD-1 and anti-CTLA-4 antibodies, implicating a synergistic effect between PI3K-AKT-mTOR pathway inhibitors and immune checkpoint inhibitors [78].

Finally, Alissafi et al. showed that CTLA-4-mediated PI3K/Akt/mTOR activation in dendritic cells (DC) led to decreased autophagy, impairing antigen presentation and T cell activation. Therefore, the authors suggested a CTLA-4-related mechanism of autophagy modification in DCs. Specifically, they described the possible attachment of Foxp3+ regulatory T lymphocytes (Tregs) to antigen-presenting DCs through CTLA-4, resulting in autophagy alterations. After treating human DCs with CTLA-4 antibodies, the formation of autophagosomes decreased, while the expression of LC-3β in rheumatoid arthritis patients was reduced, indicating decreased autophagy levels [75]. Although more studies are necessary to better understand this pathway, its modification could be a key to better anti-tumor T cell immunity.

## 5. Other Pathways of Immunoregulation and Autophagy in Breast Cancer

### 5.1. Tumor Promotion

Another study supporting the connection between autophagy and the immune microenvironment involved the expression of lysosome-associated membrane protein 2a (LAMP2a), a glycoprotein involved in autophagosome–lysosome interaction [79]. The upregulation of this protein in tumor-associated macrophages in women with breast cancer was shown to predict poor prognosis, while its inactivation enhanced macrophage tumor cytotoxicity and tumor suppression [80].

The role of autophagy in breast cancer cell-promotion through immune mechanisms was also highlighted in a study demonstrating that the epithelial–mesenchymal transition (EMT) indirectly activated autophagy and impaired CTL-mediated lysis [81]. Additionally, hypoxia-induced autophagy was found to be involved in the resistance of breast cancer cells to natural killer cell-mediated lysis (NKC-lysis; Figure 1) [82].

**Table 1 metabolites-12-00966-t001:** Clinical trials evaluating monotherapy of HCQ/CQ or combined with anti-tumor drugs in breast cancer.

ClinicalTrials.gov ID	Intervention	Study Phase	Location	Status	Start Date	Completion Date	Participants	Condition	Details	Primary Outcome	Results
NCT00765765	HCQ + Ixabepilone	Phase 1–2 Non-randomized Open label	USA	terminated	February 2009	December 2011	6	Metastatic Breast Cancer	Dose escalation from 200 mg po qd to 200 mg po bid	Assess the antitumor activity, measured by tumor response rate, in patients who receive this regimen as a third-line treatment	The study was closed early due to slow accrual. Insufficient data were collected to analyze this outcome measure
NCT01292408	HCQ	Phase 2 Open Label	Nether-lands	Unknown	January 2011	-	20	Breast Cancer	Included patients with core-biopsy proven invasive adenocarcinoma of the breast	Detect differences in endogenous hypoxia markers (CA9, PAI-1, VEGF) and autophagy (LC3b) before and after treatment with HCQ.	-
NCT01446016	CQ + Taxan/ Taxotere/ Abraxane/ Ixabepilone	Phase 2 Non-randomized Open label	USA	Completed	Sept 2011	March 2019	47	Advanced or Metastatic Breast Cancer	CQ in combination with Taxane or Taxane-like chemo agents in the treatment of patients with advanced or metastatic breast cancer who have failed anthracycline chemo base therapy	To determine the anti-tumor activity of the combination of CQ + Taxane or Taxane-like chemo agents (Paclitaxel, Docetaxel, Abraxane, Ixabepilone) measured by overall response rate	The overall response rate was 45.16%, the combination was well tolerated with only 13.15% of patients experiencing Grade ≥ 3 adverse events.
NCT02333890	CQ vs. Placebo (prior to surgery)	Phase 2 Randomized double-blind placebo-controlled	Canada	Completed	July 2015	March 2018	60	Invasive Breast Cancer	Included patients with newly diagnosed histologically confirmed primary invasive breast cancer whowere not undergoing any treatment while awaiting surgery in the next 2–6 weeks	Effect of a brief course of CQ on tumour proliferation and apoptosis based on Ki67 and TUNEL assays	No significant differences between the CQ or placebo arms in Ki67 index pre- and post-drug treatment. Adverse effects were minimal and all classified as grade 1. The effects were significant enough to cause nearly 15% of patients to discontinue therapy
NCT01023477	CQ	Phase 1–2 Non-randomized Open label	USA	Completed	December 2009	October 2016	12	Ductal Carcinoma In Situ (DCIS)	CQ standard dose (500 mg/week) or CQ low dose (250 mg/week) for 1 month prior to surgical removal of the tumor.	Tumor response evaluated by RECIST criteria as measured by breast MRI	Measurable reduction in proliferation of DCIS lesions and enhancement of immune cell migration into the duct
NCT02414776	HCQ + hormonal therapy	Phase 1–2 Non-randomized Open label	USA	Terminated	July 2014	November 2015	3	ER+ Metastatic Breast Cancer	-	Number of Participants with Adverse Events as a Measure of the safety profile of orally administered HCQ with hormonal therapy	-
NCT04523857	Abemaciclib + HCQ vs. Abemaciclib	Phase 2 Randomized Open label	USA	Not yet recruiting	April 2021	-	66	Invasive Breast Cancer	Rate of protocol defined “severe toxicity” during cycle 1 (4 weeks) of combination HCQ 600 mg BID and Abema (at 100 mg and 150 mg BID) in a safety cohort of 6 patients at each dose of Abema	Incidence of treatment-emergent adverse events, Frequency of “clearance” of bone marrow DTCs by arm after 6 cycles of study treatment.	-
NCT03032406	HCQ + Everolimus vs. Everolimus vs. HCQ	Phase 2 Pilot Randomized	USA	Recruiting	January 2017	-	60	Breast Cancer Stage IIB	Histologically-confirmed, primary, invasive breast cancer diagnosed within 5 years of study entry.	Number of Adverse Events	-
NCT04316169	Abemaciclib + HCQ	Phase 1 Non-randomized Open Label	USA	Not yet recruiting	July 2021	-	44	HR+/Her 2- Advanced Breast Cancer	Arm A: Abemaciclib + HCQ 200 mg b.i.d. Arm B. Abemaciclib + HCQ 400 mg b.i.d. Arm C: Abemaciclib + HCQ 600 mg b.i.d. Arm D: Abemaciclib + HCQ + endocrine therapy	To determine safety and tolerability of HCQ + abemaciclib and HCQ + abemaciclib + endocrine therapy in HR+/Her2- Advanced Breast Cancer	-
NCT03774472	HCQ + Palbociclib + Letrozole	Phase 1–2 Open label	USA	recruiting	August 2018	-	54	Advanced, metastatic (stage IV) Breast Cancer (phase I) Early stage (stage I-III) Breast Cancer (phase II)	This phase I/II trial studies the side effects and best dose of HCQ when given together with palbociclib and letrozole before surgery in treating participants with estrogen receptor positive, HER2 negative breast cancer.	Incidence of adverse events, Change in breast tumor proliferation index (Ki67), Change in autophagy, Change in senescence, Change in cell cycle control, Change in proportion of patients achieving tumoral complete cell cycle arrest	-

**Table 2 metabolites-12-00966-t002:** Preclinical in vivo studies evaluating the combined use of anti-tumor drugs with HCQ or CQ in breast cancer.

Studies	Anti-Tumor Medication	Autophagy Inhibitor	Tumor Cells	Comparison	Results
Lefort, 2014 [83]	Cyclophosphamide + Adriamycin (AC)	Chloroquine (CQ)	MDA-MB-231 human breast cancer cells	AC vs. AC + CQ	The combined group experienced an additive tumor growth inhibition of 41% compared to AC treatment alone and a reduction in lung metastases
Liang, 2016 [84]	Carboplatin	Chloroquine (CQ)	SUM159 cells breast cancer cells	Carb vs. Carb + CQ	Carb + CQ reduced tumor growth, decreased mitochondrial metabolic activity, decreased cell viability, and increased levels of LC3b-II and p62 demonstrating that CQ can successfully inhibit autophagy induced by carboplatin
Shoemaker, 1978 [85]	5-Fluorouracil (5-FU)	Chloroquine (CQ)	C3HBA breast cancer cells	5-FU + CQ vs. control, 5-FU vs. 5-FU + CQ	The 5-FU + CQ group had significantly reduced tumor size compared to control group and 5-FU group
Shoemaker, 1979 [86]	6-Propylthiouracil (PTU) + 5-Fluorouracil (5-FU)	Chloroquine (CQ)	C3HBA breast cancer cells	5-FU + PTU + CQ vs. control group	Significant reduction in tumor growth compared to control group
Loehberg, 2012 [29]	Everolimus	Chloroquine (CQ)	MCF7 breast cancer cells	Everolimus + CQ vs. control group	The combined treatment group showed significant weight (4.1-fold) and size (4.6-fold) reduction compared to control
Seront, 2013 [87]	Rapamycin	Chloroquine (CQ)	MDA-MB-231 and MCF-7 breast cancer cells	Rapamycin vs. Rapamycin + CQ vs. CQ	When combined with CQ, rapamycin did not further alter tumor progression in either model cancer cell type, suggesting that potential rapamycin-induced autophagy was not playing a critical role in these tumors. Tumor growth reduction was observed only in mice with large, hypoxic mammary tumors
Dragowska, 2013 [88]	Gefitinib	Hydroxychloroquine (HCQ)	JIMT-1 breast cancer cells	Gefitinib vs. HCQ vs. Gefitinib + HCQ	Notably, when gefitinib was used in combination with HCQ there was a significant 58% reduction in tumor volume compared to vehicle-treated controls
Cufi, 2013 [89]	Trastuzumab	Chloroquine (CQ)	JIMT-1 breast cancer cells	Trastuzumab vs. CQ vs. Trastuzumab + CQ	The tumor size in the combination group was drastically reduced in a synergistic manner compared to control and monotherapy groups
Ratikan, 2013 [90]	Radiotherapy	Chloroquine (CQ)	MCaK breast cancer cells	Radiotherapy + CQ	Chloroquine blocked radiation-induced autophagy and drove MCaK cells into a more rapid apoptotic and more immunogenic form of cell death
Thomas, 2012 [91]	Nelfinavir + Celecoxib	Chloroquine (CQ)	MDA-MB-468 and MCF-7 breast cancer cells	Nelfinavir + Celecoxib + CQ	Synergistic enhancement of tumor cell killing by ERSA compounds, particularly in triple-negative breast cancer (TNBC) cells.

**Table 3 metabolites-12-00966-t003:** Important Clinical Trials of Checkpoint Inhibitors in Breast Cancer (BC).

ClinicalTrials.gov ID	Intervention	Study Phase	Condition	Sample Size	Completion Year	OS (Median)	PFS (Median)	ORR
KEYNOTE-012 (NCT01848834)	Pembrolizumab	Phase Ib	Metastatic PD-L1 + TNBC	32	2016	11.2 mo	1.9 mo	16%
KEYNOTE-086 (NCT02447003)	Pembrolizumab	Phase II	Advanced PD-L1 + TNBC	170	2019	9 mo	2 mo	5%
KEYNOTE-028 (NCT02054806)	Pembrolizumab	Phase Ib	Metastatic PD L1 + BC	25	2021	8.6 mo	-	12%
KEYNOTE-150 (NCT02513472)	Pembrolizumab + Eribulin mesylate	Phase Ib/II	Metastatic TNBC with or without previous chemotherapy	167	2019	16.1 mo	4.1 mo	23%
TOPACIO (NCT02657889)	Niraparib + Pembrolizumab	Phase II	TNBC	55	2018	-	-	18.2%
PANACEA/ KEYNOTE-014 (NCT02129556)	Pembrolizumab	Phase II	HER2+ BC which has progressed on trastuzumab	52	2017	Estimated 65% at 12 mo in PD-L1+	12% at 12 mo in PD-L1 + p	15% (PD-L1 + pop)
KEYNOTE-119 (NCT02555657)	Pembrolizumab vs. Chemotherapy (capecitabine, eribulin, gemcitabine, vinorelbine)	Phase III	Metastatic TNBC	622	2019	9.9 vs. 10.8 mo	2.1 vs. 3.4 mo	18% vs. 9%
KEYNOTE-355 NCT02819518	Pembrolizumab + Chemotherapy vs. Placebo + Chemotherapy	Phase III	TNBC	882	2021	-	7.5 vs. 5.6 mo	-
NCT01375842	Atezolizumab	Phase I	Advanced TNBC	116	2018	17.6 mo	1.4 mo	24% in 1st-line treatment, 6% > 1 previous treatments
NCT01633970	Atezolizumab + Nab-Paclitaxel	Phase Ib	TNBC (stage IV or locally recurrent)	33	2020	14.7 mo	5.5 mo	39%
KATE2 (NCT02924883)	Trastuzumab emtansine + Atezolizumab vs. Trastuzumab emtansine + Placebo	Phase II	HER2+ Locally Advanced/Metastatic BC with Prior Trastuzumab and Taxane Based Therapy	202	2017	-	8.2 vs. 6.8 mo	46% vs. 44%
IMpassion130 (NCT02425891)	Atezolizumab + Nab-paclitaxel vs. Placebo + Nab-paclitaxel	Phase III	Metastatic TNBC	910	2019	21 mo vs. 18.7 mo	7.2 mo vs 5.5 mo	56% vs. 46%
IMpassion131 (NCT03125902)	Atezolizumab + Paclitaxel vs. Placebo + Paclitaxel	Phase III	TNBC (advanced or metastatic)	651	2019	18.1 vs. 22.8 mo	5.7 vs. 5.6 mo	49% vs. 41%
JAVELIN (NCT01772004)	Avelumab	Phase I	Metastatic Breast Cancer (MBC)	168	2019	8.4 mo	1.4 mo	3%
TONIC (NCT02499367)	Nivolumab + Radiation therapy/Doxorubicin/Cyclophosphamide/Cisplatin	Phase II	TNBC	66	Active, not recruiting	-	-	20%

ORR: objective response rate; OS: overall survival; PFS: progression-free survival.

### 5.2. Tumor Suppression

As with other cancers, autophagy also has a potential role in breast cancer suppression through immune activation. Autophagy activation has been reported to promote CTL- and NKC-mediated lysis of breast cancer cells [92,93,94]. In another study on human breast cancer cells, the absence of microtubule-associated protein 1 light chain 3 β (LC-3β) in the cytoplasm (which indicates reduced autophagy) and the loss of nuclear HMFB1 expression (which indicates reduced inflammatory response by immune cells) influenced the composition of immune infiltrates into the tumor. This improved the immunosurveillance profile of the tumor and improved overall and metastasis-free survival [95].

### 5.3. “Triple Negative” Breast Cancer (TNBC)

Autophagy was shown to be necessary for survival in nutrient-rich environments for some TNBC subtypes. In these cancer cell lines, autophagy led to paracrine secretion of IL-6, a pro-inflammatory cytokine, leading to increased signal transducer and activator of transcription (STAT3) phosphorylation and breast cancer stem-cell proliferation [95,96]. This was further supported by evidence that, even when autophagy was inhibited, IL-6 administration led to cancer stem-cell maintenance. Paradoxically, IL-6 had the opposite role in nutrient-poor environments: autophagy resulted in decreased IL-6 secretion, which led to increased breast cancer stem-cell maintenance.

The restriction of tumor aerobic glycolysis in two TNBC mouse models was found to inhibit the expression of tumor granulocyte colony-stimulating factor (G-CSF) and granulocyte macrophage colony-stimulating factor (GM-CSF) through a complex molecular network involving the autophagy pathway. This led to augmentation of T cell immunity, immune-mediated tumor reduction, reduced metastasis, and prolonged survival [97]. These results highlight the need to specify the context of autophagy and immunoregulatory mechanisms for prognostic and therapeutic purposes.

## 6. Autophagy, Immunotherapy, and Other Types of Treatment in Breast Cancer

### 6.1. Hormonal Therapy

The mechanistic studies mentioned above suggest that autophagy could impact the efficacy of immunotherapy, which has been demonstrated to a limited extent in breast malignancies. In breast cancer metastasized to the lungs, metformin increased the sensitivity to immunotherapy with synthetic cytosine phosphate-guanosine (CpG) oligodeoxynucleotides. These are short single-stranded DNA molecules containing unmethylated CpG dinucleotides in particular sequences that may induce autophagy activation leading to an increased CTL response [98]. In another study highlighting the role of autophagy in tumor suppression, immunotherapy with a Toll-Like receptor-5 (TLR5) agonist directly increased autophagy and decreased breast cancer proliferation via increased activity of MAP1S [99]. This is an adaptor protein in the autophagy pathway that participates in microtubular coordination and regulation of autophagy-mediated suppression of tumorigenesis [100].

### 6.2. Chemotherapy

Immune-mediated cell death and autophagy might also be indirectly linked to the use of conventional chemotherapeutic agents. Vinorelbine inhibits mitosis by interacting with tubulin and is a potential treatment of inflammatory breast cancer. It has been shown to have an anti-tumor effect by inducing a robust inflammation via Toll-Like receptor-4 (TLR4) activation, cytokine induction, and cell death via mitotic apoptosis and autophagy. Further studies are needed to confirm whether TLR4-activating molecules or immune-checkpoint inhibitors could augment the anti-tumor actions of vinorelbine [101].

### 6.3. Radiation Therapy

Other studies have highlighted that autophagy perturbation itself could be considered as immunotherapy. Radiation sensitization was improved through autophagy inhibition, likely involving a shift to increased MHC Class I expression followed by a subsequent increase in CTL activity [90]. Although each of the above studies highlights the impact of autophagy in breast cancer immunotherapy and are promising therapeutic strategies, they also have limited generalizability because each used a different immunotherapy regimen (e.g., irradiated tumor cells, TLR5 activation, reactive T-cells, IFN-γ) in the setting of different breast cancer subtypes. Furthermore, the experimental method to perturb autophagy could also impact the results. One study observed a change in immunotherapy by transient exogenous autophagy inhibition with chloroquine, but there was decreased sensitivity with cell-intrinsic autophagy knockdown [102]. Taken together, the studies highlight the often paradoxical roles of both autophagy and immune regulation in breast cancer, as well as the implications for the ultimate implementation of successful immunotherapy strategies.

## 7. Conclusions

Understanding the interactions between immune checkpoint molecules and autophagy and their potential utilization to treat women with a breast malignancy is at an early stage of development. Additional investigations are needed to identify agents and their optimal utilization to promote anti-tumor cell immunity while not interfering with self-tolerance mechanisms. Studies are needed to assess the efficacy of potential agents in treatment of the different classifications of breast cancer at the histological and molecular level, as well as to determine their efficacy in conjunction with the therapeutic guidelines that already exist in breast cancer treatment.

**Figure 1 metabolites-12-00966-f001:**
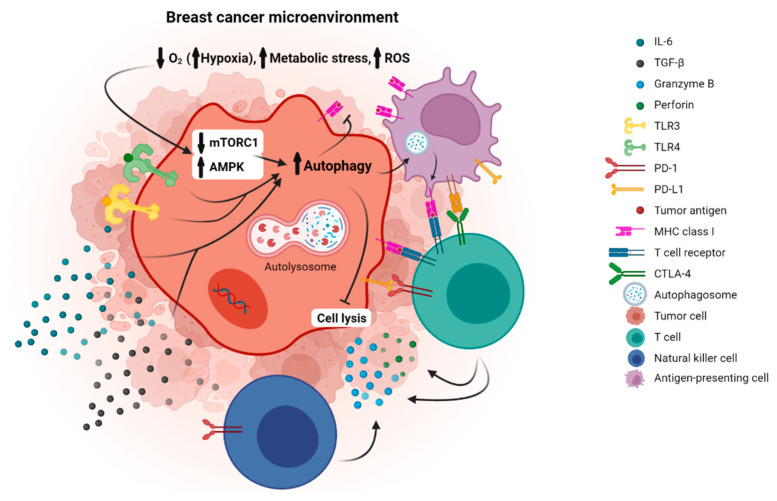
This figure summarizes pathways and molecules that contribute to breast cancer cell survival through autophagy induction. The increased metabolic stress, hypoxia, and ROS (reactive oxygen species) characterizing the cancer cell microenvironment leads to the inhibition of AKT and mammalian target of rapamycin (mTOR), resulting in autophagy induction. The latter is associated with autophagy-mediated antigen degradation and inhibition of DCs activity, impairing T cell-mediated killing and promoting tumor growth [103,104,105].

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
