# Peer review of "Targeting Both Autophagy and Immunotherapy in Breast Cancer Treatment"

_metabolites, 2022, doi:10.3390/metabo12100966_

Round 1

Reviewer 1 Report

Dear authors,

This review manuscript (Ms#metabolites-1928269) entitled “Targeting both autophagy and immunotherapy in breast cancer treatment” by Giannopoulos et al. describes the current information regarding the relevance of tumor formation and malignancy with autophagy and immune check-point mechanisms in breast cancer. I’ll provide you with a few comments below.

Comment-1 (Line 148): “Major Histocompatibility Complex” à “major histocompatibility complex”

Comment-2 (Lines 207-208): “triple negative breast cancer (TNBC)” à “TNBC”

Comment-3 (Lines 241-242): The authors described, “According to these findings, PD-L1 is an mTOR activator.” However, Zhang et al. reported that the mTORC1/2 inhibitor decreased PD-L1 expression in human non-small cell lung cancer (ref.64). The authors seem to misunderstand ref#64. 

Comment-4 (Line 266): “…the protein, LC-3β, that…” à “…the protein, microtubule-associated protein 1 light chain 3 β (LC-3β), that…”

Comment-5 (Line 304): “breast cancer tumor cell” à “breast cancer cell”

Comment-6 (Lines 321-322): “Signal Transducer and Activator of Transcription” à “signal transducer and activator of transcription”

Author Response

Comment-1 (Line 148): “Major Histocompatibility Complex” à “major histocompatibility complex”

The manuscript has been edited accordingly.

Comment-2 (Lines 207-208): “triple negative breast cancer (TNBC)” à “TNBC”

Changes have been implemented in the manuscript.

Comment-3 (Lines 241-242): The authors described, “According to these findings, PD-L1 is an mTOR activator.” However, Zhang et al. reported that the mTORC1/2 inhibitor decreased PD-L1 expression in human non-small cell lung cancer (ref.64). The authors seem to misunderstand ref#64. 

We would like to thank the reviewers for allowing us to clarify this statement. Specifically, Zhang et al. studied how mTOR changes affect PD-L1 expression. Their experiments showed that mTOR inhibition downregulates PD-L1 expression in NSCLC cells. As such, the authors concluded that the mTOR pathway could mediate the expression of PD-L1. Along those lines, we have edited the manuscript to better present the results of Zhang et al.

Comment-4 (Line 266): “…the protein, LC-3β, that…” à “…the protein, microtubule-associated protein 1 light chain 3 β (LC-3β), that…”

 We edited the manuscript according to this suggestion.

Comment-5 (Line 304): “breast cancer tumor cell” à “breast cancer cell”

“Breast cancer tumor promotion” has been changed to ‘breast cancer cell promotion”

Comment-6 (Lines 321-322): “Signal Transducer and Activator of Transcription” à “signal transducer and activator of transcription”

 We edited the manuscript according to this suggestion.

Reviewer 2 Report

This is a comprehensive and well written review which is summarizing the combination treatment of autophagy inhibitors and checkpoint inhibitors for the treatment of breast cancer.

The topic of such a combination was previously documented but it is indeed important to stress the point that such combination treatments are the future of cancer therapy, and other treatment modalities will also be engaged.

Gao, W., Wang, X., Zhou, Y. et al. Autophagy, ferroptosis, pyroptosis, and necroptosis in tumor immunotherapy. Sig Transduct Target Ther 7, 196 (2022). https://doi.org/10.1038/s41392-022-01046-3

Ying-Hua Guan, Na Wang, Zhen-Wei Deng, Xi-Guang Chen, Ya Liu,

Exploiting autophagy-regulative nanomaterials for activation of dendritic cells enables reinforced cancer immunotherapy. Biomaterials, Volume 282, 2022, 121434, ISSN 0142-9612,

https://doi.org/10.1016/j.biomaterials.2022.121434.

Duan, Y., Tian, X., Liu, Q., Jin, J., Shi, J., Hou, Y. Role of autophagy on cancer immune escape. Cell Commun Signal 19, 91 (2021). https://doi.org/10.1186/s12964-021-00769-0

Author Response

This is a comprehensive and well written review which is summarizing the combination treatment of autophagy inhibitors and checkpoint inhibitors for the treatment of breast cancer.

The topic of such a combination was previously documented but it is indeed important to stress the point that such combination treatments are the future of cancer therapy, and other treatment modalities will also be engaged.

We agree that this topic has been previously documented in the literature. However, our review focuses specifically on breast cancer in an effort to redirect clinicians’ and researchers’ interest in this area. Therefore, we present all current clinical and pre-clinical data on the impact of autophagy regulation  on breast cancer treatment. Additionally, we justify the possible synergistic effect of the combination of autophagy modulators and checkpoint inhibitors. We believe that this approach could lead the way to more efficient breast cancer treatment and benefit a large patient population.

Reviewer 3 Report

Authors reported a review article entitled “Targeting Both Autophagy and Immunotherapy in Breast Cancer Treatment” is complete and well within the scope of this journal. This MS reviewed the Immunotherapy and autophagy in breast cancer treatment, where both shown to have intertwining roles in cancer. Further, they explore the interactions between autophagy and two clinically relevant immune checkpoint pathways; the programmed cell death-1 receptor with its ligand (PD-L1)/PD-1 and the cytotoxic T-lymphocyte–associated protein 4 (CTLA-4)/CD80 and CD86 (B7-1 and B7-2). This manuscript is well-organized and interesting to the reader. However, some minor issues that need to be addressed before acceptance.

Minor Comments

1.     The format of the references given in the text is inconsistent. The author must adhere to the standard structure for citing references in the body of the manuscript. Ex. Line 22, 155, 177, 186, etc.,

2.     In vivo and in vitro should be italicised throughout the manuscript.

3.     Each section should have a list of recent references. The author should consider citing works from the past five years.

Author Response

Minor Comments

The format of the references given in the text is inconsistent. The author must adhere to the standard structure for citing references in the body of the manuscript. Ex. Line 22, 155, 177, 186, etc.,

Thank you for this comment! We have changed the references format so that it is reported consitently throuhgout the manuscript.

In vivo and in vitro should be italicised throughout the manuscript

The suggested changes have been implemented in the manuscript.

Each section should have a list of recent references. The author should consider citing works from the past five years.

We included the most recent studies throughout the manuscript; however, when we referred to important findings that were necessary for the flow of the manuscript, we cited the initial study that demonstrated these results. Older studies were only included if their findings were still supported from the more recent literature.